# Genetics and Sport Injuries: New Perspectives for Athletic Excellence in an Italian Court of Rugby Union Players

**DOI:** 10.3390/genes13060995

**Published:** 2022-06-01

**Authors:** Maria Elisabetta Onori, Massimo Pasqualetti, Giacomo Moretti, Giulia Canu, Giulio De Paolis, Silvia Baroni, Angelo Minucci, Christel Galvani, Andrea Urbani

**Affiliations:** 1UOC di Chimica, Biochimica e Biologia Molecolare Clinica, Fondazione Policlinico Universitario A. Gemelli I.R.C.C.S, Via della Pineta Sacchetti 217, 00168 Rome, Italy; massimo.pasqualetti@yahoo.com (M.P.); giacomo.moretti@policlinicogemelli.it (G.M.); giuliacanu@gmail.com (G.C.); giulio.depaolis83@gmail.com (G.D.P.); silvia.baroni@unicatt.it (S.B.); angelo.minucci@policlinicogemelli.it (A.M.); andrea.urbani@policlinicogemelli.it (A.U.); 2Dipartimento di Scienze Biotecnologiche di Base, Cliniche Intensivologiche e Perioperatorie, Università Cattolica del Sacro Cuore, Largo Francesco Vito 1, 00168 Rome, Italy; 3Department of Laboratory Medicine and Pathology, AUSL/AOU Ospedale di Baggiovara, Via Giardini 1355, 41126 Modena, Italy; 4Laboratorio Analisi Cliniche, Ospedale Pediatrico Bambino Gesù, Piazza di Sant’Onofrio, 4, 00165 Roma, Italy; 5Laboratorio di Scienze Dell’esercizio Fisico e Dello Sport, Dipartimento di Psicologia, Università Cattolica del Sacro Cuore, Viale Suzzani 279, 20162 Milan, Italy; christel.galvani@unicatt.it

**Keywords:** rugby union, *ACE*, *ACTN3*, *COL1A1*, *MCT1*, injuries

## Abstract

Several genes are involved in sport performance, especially in injuries incidence. The aim of this study was to investigate the association of *ACE*, *ACTN3*, *COL1A1*, and *MCT1* genotypes and injuries in rugby players in order to find a genotype/phenotype correlation and provide useful information improving athletic performance. One-hundred male professional and semiprofessional rugby players were selected. Analysis was performed genotyping the genes *ACE*, *ACTN3*, *COL1A1*, and *MCT1* as candidate gene of interest involved in athletic performance. A control group of non-athletic Italian male participants was analyzed to compare the results. We found statistical significance of *MCT1* rs1049434 AA for total injuries (χ^2^ = 0.115; *p* = 0.003) and bone injuries (χ^2^ = 0.603; *p* = 0.007) in the rugby athlete population. No statistical significance was found between injury incidence and *ACE*, *ACTN3*, *COL1A1* genotypes. The *MCT1* AA genotype is associated with the incidence of total and bone injuries in the rugby player population. Although environmental factors such as lifestyle, diet, training, and stress can influence athletic performance, our data demonstrated the importance of genetic study in sport aimed at developing personalized training and achieving the best possible athletic excellence.

## 1. Introduction

Sport-specific performance tests provide useful information to improve athlete performance at different levels, but more importantly to get a clearer picture of injury risk for each athlete. Several studies examined the incidence of injury in different contact sports [1]. Rugby union (RU) is an intermittent running-based team sport with high frequency of physical contacts [2] and its injury rate is higher than other team sports (but comparable to other collision sports) [3]. For a rugby athlete, injuries are one of the most disabling factors. Direct (traumatic) injuries result from collisions or physical impacts during sporting activities, while indirect (non-traumatic) injuries are caused by systemic failure due to overexertion during practice [4] such as a structural muscle injury or inappropriate warm-up [5]. Inadequate recovery after training and working beyond the pain threshold may be other factors contributing to the injury aggravation. 

Different studies demonstrated genomic involvement in sports performance and the importance of correlation between genomics and sport excellence [6,7]. The correlation between genetics and sport susceptibility may provide information to improve athletic performance, including injury risk reduction [8]. Genetic component may explain approximately 66% of the variance in athlete status depending on sport discipline [9]. In fact, genotype-phenotype correlation could provide decisive information to guide the athlete toward better sports performance [10]. 

Angiotensin converting enzyme (*ACE*) gene encodes the enzyme responsible for the conversion of angiotensin I into form II expressed in various tissues including skeletal muscles. *ACE I/D* (rs1799752) involves an insertion (I) or a deletion (D) of a 287 bp Alu sequence in intron 16 of the gene, generating three different genotypes: II (Insertion in homozygosity), ID (heterozygosity of Insertion/Deletion), DD (Deletion in homozygosity). Blood level of *ACE* activity in people with DD genotype is twice as high as those with genotype II, while blood *ACE* activity is intermediate in people with ID genotype. I allele is associated with lower serum and tissue *ACE* activity and better performance in endurance sports, whereas D allele shows that its increased circulatory and tissue activity improve performances in power or sprint sports [11]. 

Alfa-actinin-3 (*ACTN3*) gene encodes α-actininin-3 protein typically expressed in rapidly contracting muscle fibers. R577X (rs1815739) genotype involves a substitution of the amino acid arginine with a stop codon. *ACTN3* XX genotype is associated with a lower percentage of fast-twitch fibers [12], while *ACTN3* RR genotype has been associated with athletic power/performance [13]. X allele carries exhibit an increases risk of general sports injuries, whereas R allele carriers show a greater increase in strength and power following high-load endurance training [14]. 

Collagen Type I α 1 Chain (*COL1A1*) gene codes for the α1 chain of Col type I, which is responsible for the high tensile strength of tendons and ligaments [15]. Several studies indicated an association between the *COL1A1* TT (rs1800012) genotype and a lower risk of soft tissue injuries such as cruciate ligament ruptures, Achilles’ tendinopathy and rupture, shoulder dislocation [16,17]. 

Monocarboxylate transporter 1 (*MCT1*) mediates the rapid transport of lactate, especially in skeletal muscle, where rapidly glycolytic muscle fibers produce high amounts of lactate. *MCT1* (rs1049434) T1470A genotype is associated with symptomatic deficiency in lactate transport: the T allele correlates with a 35–40% reduction in lactate transport rate compared to the A allele [18]. 

Basing on the results of previous studies that correlated genes with the risk of injuries, we aimed to investigate *ACE*, *ACTN3*, *COL1A1*, and *MCT1* as candidate genes of interest in order to evaluate a possible association between genotypes and the injury risk in an Italian cohort of rugby union players.

## 2. Materials and Methods

### 2.1. Participants and Study Design

A total of 100 male professional and semi-professional rugby players from Lazio Rugby 1927 and from G.S. Fiamme Oro Rugby of Polizia di Stato were enrolled. Except for 10 participants, the other participants were Italian (Caucasian). All athletes included in the study based on medical history and injury data, while athletes with personal and family history related to osteoporosis, collagenopathies and tendinitis were excluded. Finally, injuries were stratified according to the type of tissue involved: bone, muscle and tendon/ligament. Contusion and laceration were excluded from the study. 

As control group, 100 Italian male no-athletic participants have been selected. All participants were older than 18 years and provided written informed consent. The present study is part of the investigation “Anthropometric and molecular profile of professional and semiprofessional athletes” and the study protocol was approved by the Ethics Committee of the Università Cattolica del Sacro Cuore in Rome (ID: 1858) in accordance with Declaration of Helsinki for Human Research. In Figure 1 (A/B) we provided an overview of the study design. 

### 2.2. Sample Preparation and DNA Analysis

All sample were analyzed at the Fondazione Policlinico Universitario Agostino Gemelli IRCCS in Rome, Italy. Blood or buccal swab samples were collected for each participant. Genomic DNA was extracted from blood or buccal swab samples using the automated MagCore Nucleic Acid Extractor (RBC Bioscience Corp., New Taipei City, Taiwan) instrument, in association with MagCore^®^ Genomic DNA Whole Blood Kit. DNA concentration was determined using the NanoPhotometer P-Class (Implen, Westlake Village, CA, USA). Analysis by PCR was conducted using primers showed in Table 1. PCR and Sanger sequencing were used for *MCT1* and *COL1A1* genotyping. In detail, PCR products were sequenced on the 3500 Genetic Analyzer (Applied Biosystems, Foster City, CA, USA) in association with SeqScape™ software. For *ACE* (I/D) genotyping, PCR and 4% electrophoresis gel were used. PCR/RFLP fragments analysis was used for *ACTN3* R577X genotyping. *ACE* and *ACTN3* genotype assignment was determined using the Invitrogen™ iBright™ CL1500 Imaging System according to the expected amplicon patterns.

### 2.3. Statistical Analysis and Data Interpretation

In population genetics studies, Hardy-Weinberg equilibrium is used for allele and genotype frequencies. We used the Statistical Package for Social Science (SPSS) for Windows version 22.0 (Statistical Package for Social Science, Chicago, IL, USA) in order to verify the correlation between injuries and polymorphism analyzed. Pearson’s Chi-square (χ^2^) tests was performed to compare genotype and allele frequencies between athletes and controls. The unpaired t-test was applied to compare injured and non-injured players for various genotypes using the dominant model of each single gene. The *p*-value for significance was set at *p* < 0.05. 

## 3. Results

In the athlete group, we related each genotype with the data injuries (Appendix A). We reported a total of 573 injuries as follows: 129 concerning muscle tissue, 247 to the bone tissue and 170 as tendon injuries. Fifteen athletes reported no injuries during their competitive career. Statistical analysis was conducted in order to verify genotype-injuries association. For the athletes’ group, we compared gene individually with total injury and with each injury type (bone, muscle and tendon/ligament). No statistical significance was found between injury incidence and *ACE*, *ACTN3*, *COL1A1* genotype (*p* > 0.05). However, a statistical significance between *MCT1* AA genotype with total injuries (χ^2^ = 0.115; *p* = 0.003) and bones injuries (χ^2^ = 0.603; *p* = 0.007) was found, showing a possible increased susceptibility to injury for athletes carrying the AA genotype. Thus, our results showed a correlation between the *MCT1* AA genotype and sport injuries, particularly bone injuries, in rugby athletes. The Figure 2 shows the genotype distribution in the rugby athletes and in the controls. 

The genotype distribution in the athletes’ population resulted as follows: *ACE* rs1799752 (30% DD, 49% ID, 21% II); *ACTN3* rs1815739 (30% CC, 48% CT, 22% TT); *COL1A1* rs1800012 (56% GG, 40% GT, 4% TT); *MCT1* rs1049434 (33% AA, 56% AT, 11% TT). For the control population: *ACE* rs1799752 (47% DD 38% ID, 15% II); *ACTN3* rs1815739 (54% CC, 31% CT, 15% TT); *COL1A1* rs1800012 (64% GG, 36% GT, 0% TT); *MCT1* rs1049434 (27% AA, 59% AT, 14% TT). 

No statistically significant difference was detected for the allele frequency between the two groups (*p* > 0.05). The genotype distribution was in Hardy-Weinberg equilibrium (*p* < 0.05).

## 4. Discussion

This study showed the potential susceptibility to different types of injuries (muscle, tendon and bone) in the court of 100 athletes of Lazio Rugby 1927 and G.S. Fiamme Oro Rugby. Biostatistical calculations were performed for all individual genes on both total lesions and individual stratifications (bone, muscle, and tendon). *ACE*, *ACTN3*, and *COL1A1* genes showed no significant associations for any type of injury (*p* > 0.05). For *MCT1* gene, we found no significant results between genotype and tendon-muscle injuries, while some studies reported an association between *MCT1* AA genotype and the higher incidence of muscle disorders in sportive players [19,20,21]. In contrast, our results showed a statistical correlation of the *MCT1* AA genotype with total injuries (χ^2^ = 0.115; *p* = 0.003) and with bones injuries (χ^2^ = 0.603; *p* = 0.007). We emphasize that our results could provide important molecular information about the possible protective role of *MCT1* TT genotype with respect to certain types of injuries. In fact, during training anaerobic glycolysis increases the concentration of lactate at the muscle level. Lactate is the result of a clinical manifestation of body fatigue during high-intensity exercise. Lactate transport across the plasma membrane is primarily mediated by monocarboxylate receptor MCTs, including *MCT1*. *MCT1* is ubiquitously expressed in lactate consuming cells and it is involved in transport of lactate from rapidly contracting muscle fibers [22]. Lactate accumulation could be considered one of the events contributing to the aggravation of physical injuries in athletes and it could be the cause of long periods of physical inactivity associated with significant psychological, physical, and economic consequences. *MCT1* AA genotype is associated with faster lactate transport rate than *MCT1* TT genotype. Thus, our study could help explain the role of increased susceptibility of *MCT1* AA genotype with injuries, especially bone lesions. In support of our results, data from literature indicate that *MCT1* is also involved in the osteoblast differentiation; particularly *MCT1* appears to be a negative regulator of osteoclast differentiation of bone marrow macrophages (BMMs) [23]. *MCT1* has been shown to mediate passive bone resorption induced by lactic acid released from the anaerobic metabolism of metastatic malignant cells [24]. In recent years, many studies have been conducted on athletes in order to establish an association between genotype and athletic performance. Research is mainly focused on identifying molecular markers related to physical performance that may provide useful information to identify athletes who have a higher predisposition to the risk of certain types of injuries (bone, muscle, or tendon injuries). Predictive genomic DNA profiling allows to understand whether a genetic advantage could lead to a more effective rehabilitation in some individuals than others [25].

Furthermore, medical and technical staff could develop personalized training programs to optimize the player’s athletic potential.

## 5. Conclusions

Recent data reported the association between *MCT1* AA genotype and higher incidence of muscle injury in sports people [19], while our data indicated a strong association between *MCT1* AA genotype and higher incidence of total and bone injuries in the group of 100 rugby players.

The study of the relationship between genetics related to injury phenotype seems to be increasingly useful in developing personalized training that allows athletes to focus on their genetic predispositions and achieve the best possible athletic performance.

Environmental elements such as training, lifestyle, and both physiological and psychological factors may cause an epigenetic change altering genes expression and affecting athletic performance. 

Our future aim is to better clarify the association of genetics and sports by Next Generation Sequencing using multigene panels in a larger population of athletes.

## Figures and Tables

**Figure 1 genes-13-00995-f001:**
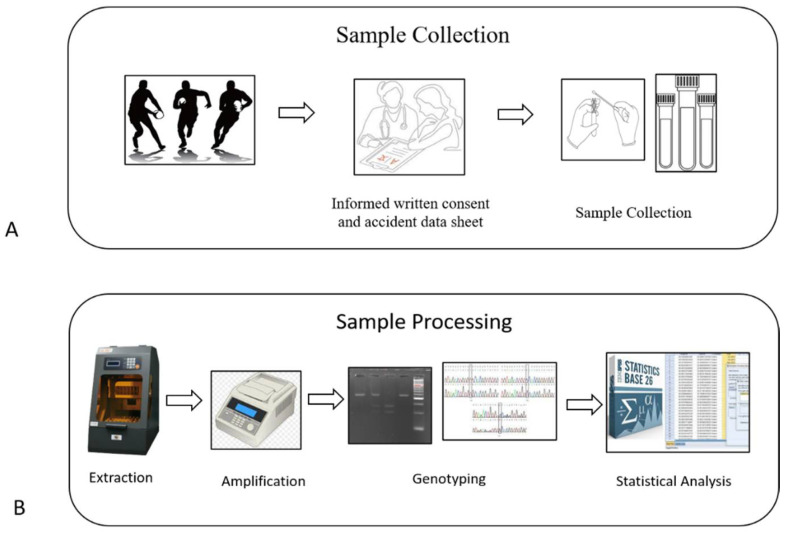
The figure shows an overview of our study design. (**A**) The figure describes the collection of sample and injuries data in rugby union elite. (**B**) The figure represents a schematic laboratory workflow for the blood sample and buccal swabs.

**Figure 2 genes-13-00995-f002:**
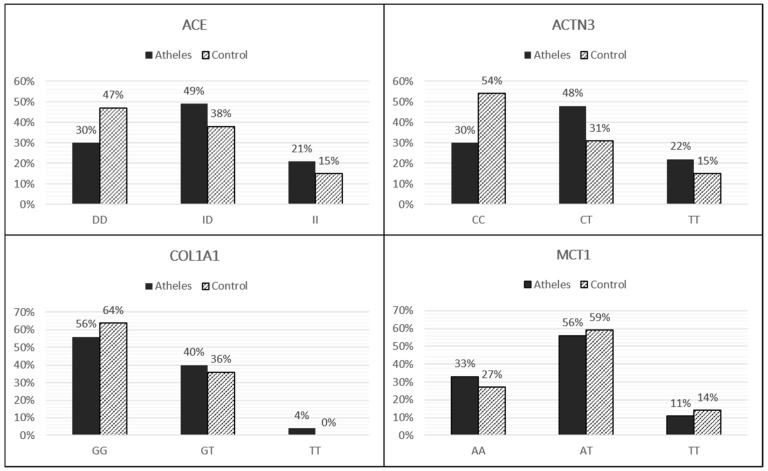
The figure shows the genotype frequency for all genes analyzed expressed as a percentage for both groups (athletes and control population).

**Table 1 genes-13-00995-t001:** Sequence of primers used for the DNA analysis.

Gene	Primers Sequence	T Annealing
*COL1A1*	Forward, 5′-GATGTCTAGGTGCTGGAGGTTAG-3′ Reverse, 5′-TGGTAGAGACAGGAGGAGGG-3′	58°
*ACE*	Forward F1, 5′-CCCATTTCTCTAGACCTGCT-3′ Forward F2, 5′-TGGGATTACAGGCGTGAT-3′ Reverse R1, 5′-AGAGCTGGAATAAAATTGGC-3′	55°
*ACTN3*	Forward, 5′-GGGCACACTGCTGCCCTTTC-3′ Reverse, 5′-GATGTCCTGCGGGCTGAG-3′	61°
*MCT1*	Forward, 5′-AGACCAGTATAGATGTTGCTGGG-3′ Reverse, 5′-CCACTGGTAGATTACAGGCCA-3′	58°

## Data Availability

The data presented in the study are available on request from the corresponding author due to restrictions (privacy).

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
