# Peer review of "Genetics and Sport Injuries: New Perspectives for Athletic Excellence in an Italian Court of Rugby Union Players"

_genes, 2022, doi:10.3390/genes13060995_

Round 1

Reviewer 1 Report

Many thanks for the article, very interesting and something I have personally worked upon. 

A few points:

  1. line 165 - this appears to be MDPI standard text "This section may be divided by subheadings. It should provide a concise and precise description of the experimental results, their interpretation, as well as the experimental conclusions that can be drawn."
  2. Limitations to the study discussed in more detail, the rise of polygenic outcomes for example (i.e. many genotypes giving one "score" for a more accurate picture of injury risk.)
  3. There are other genes that are associated with injury risk, was there a reason for these to be excluded from the study? COL5A1 for example. 
  4. Will or could epigenetic change alter how the genes work and therefore affect injury, this doesn't have to be in any great depth but mentioning this may aid your work through futureproofing and open avenues to further study. 

Many thanks again for your work and I wish you the best in the future. 

Author Response

Dear Reviewer,

I provided a point-by-point response to the comments. I uploaded the file. Please see the attachment.

Thank you

Sincerely

Reviewer 2 Report

The manuscript under this review concerns interesting concept of interaction between ACE, ACTN3, COL1A1, MCT1 and sports injuries in 100 male rugby players and 100 volunteers. The purpose of this paper is important, however, the masuscript contains numerous methodological errors, inaccuracies, and shortcomings.

Major points:

  1. The manuscript should be checked in regard to English grammar. Native speaker should re-write many sentences/passages.
  2. Authors should check the text and improve the punctuation mistakes. The genes should be always written in italics. Authors should check the abbreviations throughout the entire manuscript. Needs correction throughout the entire manuscript.
  3. Introduction section

The paper has not adequate theoretical reflection on subject matter. What I miss most is the explanation of why the authors choose such diverse genes for their analysis. The description of genes and polymorphisms should be re-written.

Authors wrote „The SLC16A1 gene located on human chromosome 1p13.2-p12 encodes the MCT1 receptor.” but in the other part of manuscript authors wrote „MCT1 gene”.

The aim of the study should be re-written, because authors did not analyse athletic performance.

The introduction is also too long.

  1. Materials and Methods section

The description of participants should be re-written, the table with the characteristics of the study group and injuries would be valuable.

‘Subjects’ should be changed to ‘Participants’. Please do not use impersonal form in the entire manuscript.

Authors should add the full name of the ethisc committee.

The Figure 1 is not necessary.

Sanger sequencing was used for MCT1 and COL1A1 genotyping? Why?

All models of inheritance, i.e. codominant, dominant, recessive, and  verdominant should be constructed.

The analysis of gene-gene interaction would be much more interesting.

  1. Results section

Lines 157 and 164 are repeated in Figure 2.

Lines 165 -167 should be removed.

Description of the Figure 2 is missing.

Tables with obtained results are missing.  

  1. Discussion section

This section is too short.

Authors wrote only about MCT1, the part explantating other results is missing. Authors should focus on the potential biochemical and physiological mechanisms behind the potential associations (also these statistically insignificant).

Author Response

(The authors gave the same response as above.)

Reviewer 3 Report

Dear Authors,

I found your work very interesting.

The manuscript is well written and the results are clearly described.

Please check the attached file for my minor suggestions

Author Response

(The authors gave the same response as above.)
